# Dynamically Gated Mixture of Experts for Multi-task Reinforcement Learning

## Abstract

Multi-task reinforcement learning (MTRL) promises unique strengths against single-task RL because of generalization across tasks through parameter sharing and composition. Existing methods rely on local routing or static compositional weights in Mixture-of-Experts (MoE) without the ability to adapt to evolving temporal context. To improve the inherently temporal and dynamic systems in MTRL, we introduce a global recurrent inhibition network (GRIN) that performs dynamic gating across time, selectively modulating expert activations based on temporally accumulated context. Our formulation propagates information across time steps to preserve global activation information across the model. Notably, using the gating approach, we found statistically significant improvements over state-of-the-art MTRL methods, with an empirical **+3.7%** improvement on the Metaworld MT50 benchmark.

## 1 Introduction

Deep reinforcement learning has demonstrated the capacity to acquire complex behaviors across diverse domains Mnih et al. (2013); Gu et al. (2017); Kalashnikov et al. (2018); Lillicrap et al. (2015). The human ability to leverage similar tasks to learn to perform many tasks inspires multi-task reinforcement learning (MTRL), which offers a path toward more general agents. To enable sharing of knowledge and parameters, Mixture-of-experts and compositional parameter-sharing approaches Hendawy et al. (2023); Sun et al. (2022) have shown promise by enabling task-specific specialization within unified architectures. However, these methods typically rely on local routing or static compositional weights, limiting adaptability as temporal context evolves during an episode.

Gating mechanisms have proven effective for controlling information flow in neural networks, such as LSTMs Hochreiter & Schmidhuber (1997) and notably, to modern attention and LLM architectures Qiu et al. (2025). Recent work Qiu et al. (2025) demonstrates that dynamic, input-dependent gating applied across time enhances nonlinear expressivity and improves optimization stability. We investigate whether these benefits extend to the complex temporal dynamics of MTRL, where agents must continuously adapt behavior based on evolving task demands. We introduce a global recurrent inhibition network (GRIN) that performs dynamic gating across time steps, selectively modulating feature activations based on temporally accumulated context.

A key challenge in applying gating to model-free deep RL is that feedforward networks restrict gating inputs to preceding layers within the current forward pass, forfeiting access to global information. Our formulation addresses this by propagating activations forward across time steps, enabling the gate to leverage any activation from the previous step and preserve global context. This mechanism directly parallels cortical inhibitory neurons that integrate diverse synaptic inputs to orchestrate neural computation.

We show that in multi-task settings, global network gating with input access to Q-value and task-specific signals enhances model performance. GIN achieves 90.0% on the Metaworld MT10 benchmark, and demonstrates a statistically significant **+3.7%** improvements with our reproduced results on the challenging MT50 benchmark. These results demonstrate that temporal gating yields improvements over state-of-the-art MTRL methods, establishing dynamic gating across time as an effective mechanism for multi-task policy learning.

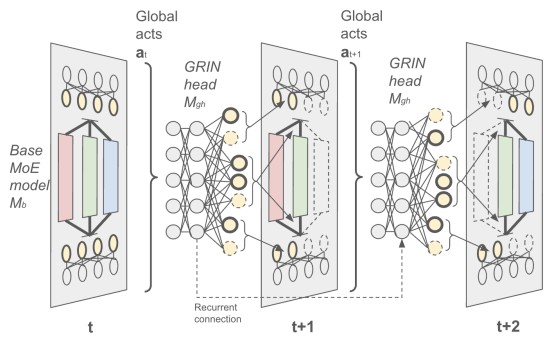
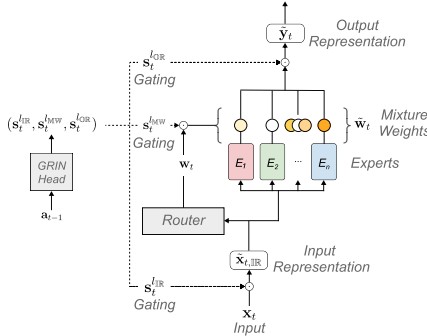

(a) Illustration of GRIN inhibition head module which takes in all global activations and dynamically changes the MoE weights and architecture.

(b) MoE preliminaries and GRIN gating locations.

Figure 1: Model Architecture

## 2 BACKGROUND AND RELATED WORK

A rich body of work in MTRL has explored diverse strategies for achieving effective generalization across tasks. The paper D'Eramo et al. (2024) established theoretical foundations showing that MTRL yields increasing benefits as the number of tasks grows, while Teh et al. (2017) propose learning individual task policies that share a common prior. The Meta-World benchmark Yu et al. (2020b) has become a standard testbed for robotic manipulation, typically trained with Soft Actor-Critic Haarnoja et al. (2018). However, naive parameter sharing can induce negative transfer when task gradients conflict. The paper Yu et al. (2020a) address this through gradient projection methods that orthogonalize task gradients, though such approaches can be sensitive to gradient variance. Modular architectures offer an alternative: Yang et al. (2020) introduce routing networks that generate task-specific parameters from a shared base model, and Devin et al. (2017) decompose policy responsibilities across robot-specific and task-specific modules. More recent work conditions shared representations on task context: Sodhani et al. (2021) learn a mixture of state encoders using task metadata, producing diverse and interpretable representations, while Perez et al. (2018) leverages feature-wise linear modulation for task conditioning.

Deep learning has designed adaptation and inhibitive mechanisms through learnable gating architectures that selectively amplify or suppress features. These appear in LSTMs (Hochreiter & Schmidhuber, 1997), Highway Networks (Srivastava et al., 2015), and Gated Linear Units (GLUs) (Dauphin et al., 2017b; Shazeer, 2020), where they prevent gradient vanishing, and enhance network sparsity. The mechanism mirrors biological cortical inhibition, where diverse inhibitory neurons modulate excitatory activity to maintain network stability and sharpen signal selectivity (Isaacson & Scanziani, 2011; Klausberger & Somogyi, 2008; Tremblay et al., 2016). Studies reveal remarkable diversity in inhibitory connectivity patterns inhibition(Gidon & Segev, 2012; Markram et al., 2004; Chini et al., 2022).

Dropout and its variants (Ba & Frey, 2013; Kingma et al., 2015; Ghiasi et al., 2018; Liu et al., 2022; Zhao et al., 2022; Li et al., 2023) demonstrate the effectiveness of stochastic adaptation during training. Capsules (Hinton et al., 2011; Sabour et al., 2017; Hinton et al., 2018) leverage dynamic routing at the unit and layer level, and remains an area of novel active research. In comparison, we focus on a global approach to improve existing ML models and MoE.

Different from prior work, we propose a global approach across the model. The propagation across time builds a global, dynamic adaptation module. This module aggregates signals from any part of the neural network and modulate MoE pathways. This design enables GRIN to provide a general mechanism for dynamic modulation. We demonstrate its effectiveness in multi-task reinforcement learning, as well as in computer vision and limited-scale language models.

## 3 ALGORITHM AND METHODS

### 3.1 PRELIMINARIES

Mixture of Experts (MoE) (Shazeer et al., 2017) architectures dynamically route inputs through specialized sub-networks. A standard MoE layer transforms an input representation ($\mathbb{IR}$) $\mathbf{x}_{\mathbb{IR}} \in \mathbb{R}^d$. In an MoE without a router model (Sun et al., 2022), $\mathbf{x}_{\mathbb{IR}}$ is transformed by the experts then aggregated by mixture weights ($\mathbb{MW}$) which could be either fixed or learned. In a gated MoE, $\mathbf{x}_{\mathbb{IR}}$ is put to a router model that computes mixture weights $\mathbf{w}_{\mathbb{MW}} = \text{Router}(\mathbf{x}_{\mathbb{IR}})$, and $\mathbf{w}_{\mathbb{MW}} = [w_1, w_2, ...]$ where $w_i$ represents the contribution of the $i$-th expert out of $N$ experts. After $\mathbf{x}_{\mathbb{IR}}$ is propagated through the expert models, an aggregation step combines expert outputs. The final output representation ($\mathbb{OR}$) is computed as: $\mathbf{y}_{\mathbb{OR}} = \sum_{i=1}^{N} w_i \cdot E_i(\mathbf{x}_{\mathbb{IR}})$, where $E_i$ denotes the $i$-th expert network and $N$ is the total number of experts. Variants include mixed MoE where all experts process inputs with soft weights from a softmax router, gated MoE that adds learnable gates to modulate expert contributions, and sparsely-gated MoE that selects only the top-$k$ experts for efficiency.

To enable precise control over network modulation, we design the GRIN head module to dynamically regulate MoE architectures. Generic MoE leverages three key representations: the input representation ($\mathbb{IR}$), the mixture weights ($\mathbb{MW}$), and the output representation ($\mathbb{OR}$). The GRIN head module generates gating masks through sigmoid non-linearity (Dauphin et al., 2017a), directly modulating these representations.

### 3.2 GRIN ARCHITECTURE WITH THE HEAD MODULE

The limitation of existing MoE approaches is that gating decisions are made locally based solely on the current input representation $\mathbf{x}_{\mathbb{IR}}$, without considering global network activation patterns or interactions between experts. We propose Global Recurrent Inhibition Networks (GRIN) that modulate the MoE architecture at three critical locations[1]—input representations ($\mathbb{IR}$), mixture weights ($\mathbb{MW}$), and output representations ($\mathbb{OR}$)—using inhibition masks ($\mathbf{s}$) derived from global network activations:

$$\tilde{\mathbf{a}}^l = \mathbf{s}^l \odot \mathbf{a}^l, \quad \text{for } l \in \{l_{\mathbb{IR}}, l_{\mathbb{MW}}, l_{\mathbb{OR}}\} \quad \text{where } \mathbf{s}^{l_{\mathbb{IR}}}, \mathbf{s}^{l_{\mathbb{MW}}}, \mathbf{s}^{l_{\mathbb{OR}}} = \sigma(\mathcal{G}(\mathbf{a}_{global})). \tag{1}$$

Here $\mathbf{a}^l$ denotes the original activation at location $l$, $\mathbf{s}^l$ denotes the inhibition mask computed by the GRIN head $\mathcal{G}$ from global activations $\mathbf{a}_{global}$ with the ending sigmoid non-linearity $\sigma$, $\odot$ denotes element-wise multiplication, and $\tilde{\mathbf{a}}^l$ denotes the gated activation. This formulation enables dynamic, globally-aware modulation that considers the full network state when making modulation and gating decisions.

As shown in Figure 1a and Figure 1b, the GRIN head model operates as a modulator and controller, taking in the global activations and producing the inhibition masks. The inhibition masks are applied to elements of the MoE model, on mixture weights ($l_{\mathbb{MW}}$) of the MoE model, which determines the weighting or selection of each expert. The other category of gating for the GRIN head is to modulate and modify the input representation ($l_{\mathbb{IR}}$) as well as the output representation ($l_{\mathbb{OR}}$) of the MoE model.

For a base model with MoE architecture, GRIN is defined as $\Gamma = (\mathcal{X}, \mathcal{M}, \mathcal{G}, \mathcal{A}, \mathcal{S}, \mathcal{L})$, where $\mathcal{X}$ denotes input data, $\mathcal{M}$ is the base MoE model, $\mathcal{G}$ is the GRIN head, $\mathcal{A}$ represents base model activations, $\mathcal{S}$ are the inhibition masks (gating masks), and $\mathcal{L}$ specifies gating target locations, and $\mathcal{L} = \{l_{\mathbb{IR}}, l_{\mathbb{MW}}, l_{\mathbb{OR}}\}$. The system evolves through discrete time steps with state representations $\mathcal{D} = \{(\mathbf{x}_t, \mathbf{a}_t, \mathbf{h}_t, \mathbf{s}_t)\}$, where $\mathbf{x}_t \in \mathbb{R}^n$, $\mathbf{a}_t \in \mathbb{R}^m$, $\mathbf{h}_t \in \mathbb{R}^k$, and $\mathbf{s}_t \in \mathbb{R}^p$.

---

[1]We define a location to be one or more layers, units or connections in the network.

---

**Algorithm 1** GRIN Forward Pass

---

**Require:** Input $\mathbf{x}_t$, Number of recurrent steps $T$, GRIN head $\mathcal{G}$, base model $\mathcal{M}$

1: **for** $iter = 1$ to $N$ **do**
2:     Initialize $\mathbf{s}_0 \leftarrow \mathbf{1}$, $\mathbf{h}_0$, $\mathbf{h}_{-1}$
3:     **for** $t = 1$ to $T$ **do**
4:         $\mathbf{h}_{t-1} \leftarrow f_{\text{RNN}}(\mathbf{h}_{t-2}, \tilde{\mathbf{a}}_{t-1})$         $\triangleright$ Update hidden state
5:         $\mathbf{s}_t^{l_{\text{IR}}}, \mathbf{s}_t^{l_{\text{MW}}}, \mathbf{s}_t^{l_{\text{OR}}} \leftarrow \sigma(\mathbf{W}_{\text{GRIN}}\mathbf{h}_{t-1} + \mathbf{b}) = \sigma(\mathcal{G}(\mathbf{h}_{t-2}, \tilde{\mathbf{a}}_{t-1}))$
6:         $\mathbf{a}_t^{l_{\text{IR}}}, \mathbf{a}_t^{\text{pretext}} \leftarrow \mathcal{M}_{\text{Pretext}}(\mathbf{x}_t)$     $\triangleright$ Forward pretext model
7:         $\tilde{\mathbf{a}}_t^{l_{\text{IR}}} \leftarrow \mathbf{s}_t^{l_{\text{IR}}} \odot \mathbf{a}_t^{l_{\text{IR}}}$        $\triangleright$ Modulate input repres.
8:         $\mathbf{w}_t \leftarrow \text{Router}(\tilde{\mathbf{a}}_t^{l_{\text{IR}}})$        $\triangleright$ Compute mixture weights
9:         $\tilde{\mathbf{w}}_t \leftarrow \mathbf{s}_t^{l_{\text{MW}}} \odot \mathbf{w}_t$        $\triangleright$ Modulate mixture weights
10:       $\mathbf{a}_t^{l_{\text{OR}}} \leftarrow \sum_{i=1}^{N} \tilde{w}_{i,t} \cdot E_i(\tilde{\mathbf{a}}_t^{l_{\text{IR}}})$     $\triangleright$ Aggregate outputs
11:       $\tilde{\mathbf{a}}_t^{l_{\text{OR}}} \leftarrow \mathbf{s}_t^{l_{\text{OR}}} \odot \mathbf{a}_t^{l_{\text{OR}}}$        $\triangleright$ Modulate output
12:       $\mathbf{a}_t^{\text{post}} \leftarrow \mathcal{M}_{\text{post}}(\tilde{\mathbf{a}}_t^{l_{\text{OR}}})$       $\triangleright$ Forward post model
13:       $\tilde{\mathbf{a}}_t \leftarrow \{\mathbf{a}_t^{\text{pretext}}, \tilde{\mathbf{a}}_t^{l_{\text{IR}}}, \tilde{\mathbf{a}}_t^{l_{\text{OR}}}, \mathbf{a}_t^{\text{post}}\}$     $\triangleright$ Aggregate activations
14:       Compute $\mathbf{loss}_t$
15:       BackProp $\mathbf{loss}_t$
16:       $\theta \leftarrow \theta + \Delta\theta$
17:     **end for**
18: **end for**

---

Figure 2: Recurrent network for GRIN with respect to activations, hidden states, and inhibition masks.

## 3.3 GRIN RECURRENT FORMULATION

GRIN employs a recurrent formulation where the inhibition masks are iteratively refined over $T$ time steps, allowing the network to observe and respond to its own activation patterns. As shown in Figure 2, at each timestep $t$, the GRIN head processes the current gated activations $\tilde{\mathbf{a}}_t$, the hidden state $\mathbf{h}_t$, and the previous hidden state $\mathbf{h}_{t-1}$ to compute updated inhibition masks $\mathbf{s}_{t+1}$, which modulate the network's forward pass. This recurrent mechanism enables the network to iteratively refine the modulation of expert weights and representations by incorporating feedback from previous choices. The state evolution is governed by the following dynamics:

$$
\begin{aligned}
\mathbf{h}_t &= f_{\text{RNN}}(\mathbf{h}_{t-1}, \tilde{\mathbf{a}}_t) \\
\mathbf{s}_{t+1} &= \sigma(\mathbf{W}_{\text{GRIN}} \cdot \mathbf{h}_t + \mathbf{b}) = \sigma(\mathcal{G}(\mathbf{h}_{t-1}, \tilde{\mathbf{a}}_t)) \\
\tilde{\mathbf{a}}_{t+1}^{l_{\text{OR}}} &= \mathcal{M}_{\text{MoE}}(\mathbf{x}_{\text{IR}} \odot \mathbf{s}_t^{l_{\text{IR}}}, \mathbf{w}_{\text{MW}} \odot \mathbf{s}_t^{l_{\text{MW}}}) \odot \mathbf{s}_t^{l_{\text{OR}}}
\end{aligned}
\tag{2}
$$

where $f_{\text{RNN}}$ can be any recurrent cell (LSTM, GRU), $\mathbf{s}_t = \{\mathbf{s}_t^{l_{\text{IR}}}, \mathbf{s}_t^{l_{\text{MW}}}, \mathbf{s}_t^{l_{\text{OR}}}\}$ are the inhibition masks at timestep $t$, and $\mathcal{M}_{\text{MoE}}(\cdot)$ denotes the modulated MoE forward pass using inhibition masks $\mathbf{s}_t$. During training, gradients flow through all timesteps via backpropagation through time, jointly optimizing the base model and GRIN head. Algorithm 1 presents the full GRIN recurrent algorithm.

## 3.4 TRAINING, OPTIMIZATION, AND INFERENCE

During training and across recurrence steps, the gradient computation graph is kept and we perform gradient back-propagation across time to optimize the recurrent neural network. We note that in the case of GRIN, when the data input is non-sequential, we choose to use the same $\mathbf{x}$ to propagate multiple time-steps. This simulates the dynamic changes in the base-model with the same data input, but across different time steps. When we use GRIN to recurrently forward propagate T steps, it is a choice to back-propagate at each time step, and sum the gradients over time. This offers a higher quality gradient for the optimization. At inference time, the pretex inhibition and GRIN models can operate on the data points sampled from the test set.

## 4 EXPERIMENTS ON MULTI-TASK REINFORCEMENT LEARNING

We evaluate the GRIN algorithm on the MetaWorld benchmark with the multi-task reinforcement learning (MTRL). MetaWorld offers a suite of reinforcement learning environments comprising up to 50 robotic manipulation tasks. In our RL model, both the actor and critic networks employ Mixture-of-Experts (MoE) architectures with orthogonalization, while the mixture-weight encoder is conditioned solely on the task-ID.

Table 1: MT10 Average Success Rate (%): comparison of GRIN with prior methods

| Method | Epoch 1 % (1M) | Epoch 2 % (2M) | Epoch 3 % (3M) | Epoch 5 % (5M) | Epoch 10 % (10M) | Epoch 15 % (15M) | Epoch 20 % (20M) |
|---|---|---|---|---|---|---|---|
| SAC (Yu et al., 2019) | 10.0±8.2 | 17.7±2.1 | 18.7±1.1 | 20.0±2.0 | 48.0±9.5 | 57.7±3.1 | 61.9±3.3 |
| MTSAC (Yu et al., 2019) | 34.9±12.9 | 49.3±9.0 | 57.1±9.8 | 60.2±9.6 | 61.6±6.7 | 65.6±10.4 | 62.9±8.0 |
| SAC+FiLM (Perez et al., 2017) | 32.7±6.5 | 46.9±9.4 | 52.9±6.4 | 57.2±4.2 | 59.7±4.6 | 61.7±5.4 | 58.3±4.3 |
| PCGrad (Yu et al., 2020) | 32.2±6.8 | 46.6±9.3 | 54.0±8.4 | 60.2±9.7 | 62.6±11.0 | 62.6±10.5 | 61.7±10.9 |
| Soft-Module (Yang et al., 2020) | 24.2±4.8 | 41.0±2.9 | 47.4±5.3 | 51.4±6.8 | 53.6±4.9 | 56.6±4.8 | 63.0±4.2 |
| CARE (Sodhani et al., 2021) | 26.0±9.1 | 52.6±9.3 | 63.8±7.9 | 66.5±8.3 | 69.8±5.1 | 72.2±7.1 | 76.0±6.9 |
| PaCo (Sun et al., 2022) | 30.5±9.5 | 49.8±8.2 | 65.7±4.5 | 64.7±4.2 | 71.0±5.5 | 81.0±5.9 | 85.4±4.5 |
| MOORE (Hendawy et al., 2024) | 36.4±7.8 | 64.4±5.5 | 72.1±6.5 | 74.8±4.0 | 80.1±6.1 | 84.8±4.3 | 88.4±3.4 |
| MOORE (**IQM**±std) | 33.5±4.3 | 65.0±4.0 | 72.8±4.0 | 74.2±3.3 | 79.8±0.4 | 84.8±4.3 | 89.5±0.3 |
| GRIN (Ours) | **45.7**±9.1 | 63.7±3.0 | 68.3±5.3 | **78.4**±5.8 | **83.0**±4.4 | **87.7**±3.5 | **89.4**±1.0 |
| GRIN (Ours, **IQM**±std) | **45.4**±5.0 | 63.3±0.4 | 69.0±1.7 | **78.3**±0.4 | **81.5**±2.6 | **89.4**±0.9 | **90.0**±0.0[4] |

Aligned with the experiment procedures of the previous work, we use the soft-actor-critic (SAC) model with a 3-layer fully connected neural network each with 400 hidden units and *tanh* non-linearity. We use 4 experts MoE model for the MT10 experiment and 6 experts MoE for the MT50 experiment. The multi-head architecture is used after the MoE. Our experiments are performed with GRIN gating on both $l_{\mathbb{MW}}$, and $l_{\mathbb{OR}}$ locations, and implemented in the MoE algorithm with task-encoder (Sun et al., 2022) and orthogonalization (Hendawy et al., 2023). The MoE architecture and GRIN are used for both Actor and Critic networks, but GRIN does not have connections across the two network. In each epoch we run the soft-actor-critic algorithm with GRIN for 100,000 iterations, with batch size of 128. For evaluation, we follow Agarwal et al. (2021) to compute the interquartile mean (IQM) of success rates across random seeds for both MT10 and MT50. Robust to outlier scores, the IQM computes the mean on the middle 50% of combined runs, after ranking the random seeds with their success rates. For the standard deviation calculation in the IQM column of the tables, the bottom and top 25% of the data are excluded. For both MT10 and MT50 comparison against MOORE Hendawy et al. (2023), we report the success rate metrics obtained by *reproducing* the authors results by running their open-source code.[2]

### 4.1 GRIN SHOWS BEST IQM RESULTS METAWORLD MT10

In Table 1, we report the evaluation success rates for MT10 in the MetaWorld environment. The mean and standard deviation of the success rate are computed across 10 random seeds. The GRIN algorithm consistently improves upon the MoE-based MTRL algorithm, despite the performance plateauing effect in MT10 also reported in (Hendawy et al., 2023). In Table 1, we also compare selected epochs with prior algorithms, including the recent PaCo (Sun et al., 2022) and MOORE approaches (Hendawy et al., 2023). We further note the MT10 success rate of $0.8923 \pm 0.0112$ reported in the recent work (Kong et al., 2025). GRIN (ours) surpasses this result, while we note that (Kong et al., 2025) reported using only 3 random seeds [3].

### 4.2 GRIN SHOWS SIGNIFICANT IMPROVEMENT OVER PRIOR ART ON METAWORLD MT50

We evaluate GRIN on the MetaWorld MT50 benchmark (MT50) with 50 distinct tasks. Our implementation builds upon the orthogonalized mixture-of-experts (MOORE) framework proposed by Hendawy et al. (Hendawy et al., 2023), employing six experts for both actor and critic networks with global activation intake and a recurrence depth of 1, as determined optimal in our ablation studies.

Table 2 presents the success rates and the IQM results comparing GRIN against MOORE, the current state-of-the-art method. Our results demonstrate that GRIN achieves substantial improvements over MOORE, with notable performance gains at 50M, 100M, and 175M environment steps. These results suggest that the global gating and modulation mechanism in GRIN effectively enhances the model's ability to handle the diverse task distribution in MT50. We note that although our reproduction results with open-source code didn't reach the reported numbers in Hendawy et al. (2023), the statistically significant improvement are clearly shown in the results. GRIN algorithm improved IQM results by **+3.7%** at 100M env. steps, shown in Table 2. This improvement sustains even as we continue to run the algorithms to 175M env. steps, as shown in Table 3.

---

[2]We used: https://github.com/AhmedMagdyHendawy/MOORE.

[3]Appendix B5 of (Kong et al., 2025)

[4]The success rate data for random seeds excluding bottom 25% and top 25% are all 90%.

| Algorithm | Env. st. | Ave. suc. | IQM |
|---|---|---|---|
| MTSAC (Yu+ 19) | 100M | 49.3±1.5% | – |
| SAC+FiLM (Perez+ 17) | 100M | 36.5±12.0% | – |
| CARE (Sodhani+ 21) | 100M | 50.8±1.0% | – |
| PaCo (Sun+ 22) | 100M | 57.3±1.3% | – |
| MOORE (Our repro.) | 50M | 51.2±2.1% | – |
| GRIN (Ours) | 50M | **55.7**±2.9% | – |
| MOORE (Our repro.) | 100M | 55.7±2.8% | 56.1±0.9% |
| GRIN (Ours) | 100M | **59.9**±1.6% | **59.8**±0.2% |

Table 2: Metaworld MT50 results.

| Algorithm | Env. st. | IQM |
|---|---|---|
| MOORE (Our repro.) | 175M | 62.1±0.5% |
| GRIN (Ours) | 175M | **63.4**±1.0% |
| GRIN (Ours) | 250M | **64.4**±1.0% |

Table 3: Extended run on MT50.

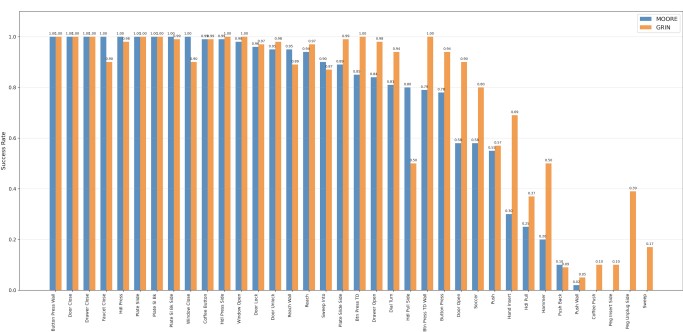

Figure 3: MT50 tasks success rate comparison GRIN compared with MOORE, reported at 50M env. steps.[5]

In Figure 3, we show the per-task performance of GRIN compared with the prior MOORE algorithm. Several tasks exhibit significant improvements. Out of 36 tasks with non-zero success rates, GRIN achieves higher success rates than MOORE on 22 tasks (61%). In contrast, MOORE outperforms GRIN on 8 tasks (17%).

### 4.3 ANALYSIS WITH OFF POLICY EVALUATION

In the MetaWorld MT10 environment, we perform off-policy evaluation by collecting 150,000 transitions in offline trajectories across tasks.

For each transition, Q values for GRIN, and MOORE as baseline, are computed with $Q = F_{\text{critic}}(s_t, a_t^*)$ where $s_t$ is the transition state, and $a_t^* = F_{\text{actor}}(s_t)$. During the forward compute, we collected the output representation ($\mathbb{OR}$) inhibition masks, the mixture weight ($\mathbb{MW}$) inhibition masks, which are used to compute average inhibition levels. Table 4 shows $\mathbb{OR}$ inhibition levels correlates the most with the Q value. In comparison the inhibition levels on mixture weights are less significant. This finding indicates GRIN's modulation may play a larger role compared to routing data in MTRL. Across data in all transitions, we fit Gaussian distributions to the baseline model's and GRIN's Q values for a Fitted Q Evaluation (FQE) analysis. Figure 4 shows the Q value improvement with GRIN is on average 8.5, and 93.5% of the samples observed improvement in Q with GRIN. Finally, we present a segment analysis. The data is segmented by the median in actor inhibition level[6]. On the top of Figure 5 we show histogram plot of Q values of transitions with more actor $\mathbb{OR}$ inhibition (above median), and on the bottom we show the plot for transitions with less. With more inhibition, we observe more improvement (+10.2) across the median of GRIN Q values vs baseline. This value is smaller (+9.5) in the opposing segment. The value distributions are visibly different across more vs less inhibition. The result shows the effectiveness and significance of modulation from the GRIN head model.

### 4.4 ABLATION STUDIES

---

[6]We compute the average across the hidden units for Actor $\mathbb{OR}$ inhibition masks, and find the median across the transitions. The median value is 0.105.

Table 4: Inh. corr. & Q improvements

| Metric | Value | Desc. |
|---|---|---|
| $\mathbb{OR}$ *Inhibition* | *Corr.* | |
| Actor $\rightarrow$ GRIN Q | $+0.47$ | +ve |
| Actor $\rightarrow$ Q impr. | $+0.31$ | +ve |
| Critic $\rightarrow$ GRIN Q | $+0.19$ | Weak +ve |
| Critic $\rightarrow$ Q impr. | $+0.14$ | Weak +ve |
| $\mathbb{IMW}$ *Inhibition* | *Corr.* | |
| Actor $\rightarrow$ Q | $-0.08$ | Slight -ve |
| Critic $\rightarrow$ Q | $-$ | Negligible |
| *Q-Value Performance* | | |
| Improved samples | $93.5\%$ | $-$ |
| Mean impr. | $8.5 \pm 7.0$ | $-$ |

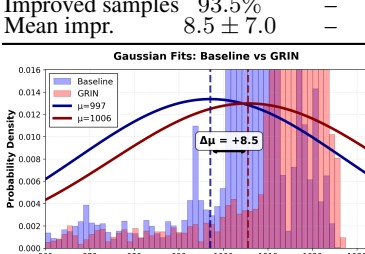

Figure 4: FQE Gaussian fit of GRIN Q values vs Baseline.

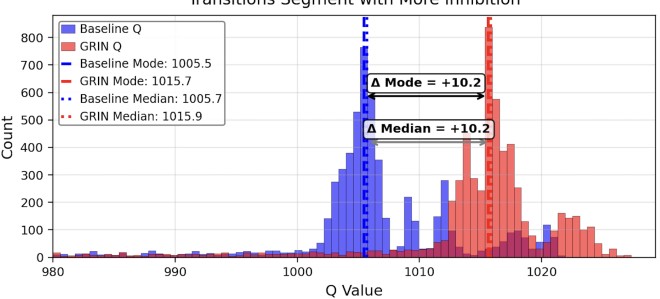

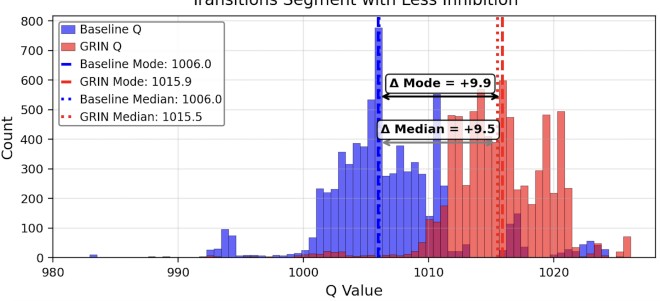

Figure 5: Segment Analysis for transitions with More $\mathbb{OR}$ inhibition (inhibition mask mean level is below the data median *0.105*) vs Less.

**Recurrent depth**. We conduct ablation studies on the Meta-World MT5 benchmark comprising five tasks to evaluate GRIN with recurrence depths ranging from 1 to 3. The recurrent implementation follows Algorithm 1. As shown in Table 5 and Figure 6, a recurrence depth of 1 yields optimal success rates indicating a single recurrence step is typically sufficient for less complex MTRL tasks. **Number of experts**. We evaluate GRIN's effectiveness by varying expert capacities across 3, 4, and 6 experts, keeping other parameters constant. Results in Figure 6 show that for the MT5 task, expanding the expert pool didn't significantly improve the results. We also perform the experiment on Minigrid Chevalier-Boisvert et al. (2023) MT7[7] shown in Table 6 and observe similar results. Although increasing the number of experts offered less improvements, the ablation aligns with results with the MT10 Q-value correlation in Table 4, and verifies that GRIN contributes more through modulation on the representations.

**Selectivity from global inputs**. Global activations make the inhibition head effective for modulation the network. While since there is a variety of base model architectures, the selection of inputs may be needed. We offer an ablation using Minigrid, which uses a convolutional net. In the base GRIN setting, we connect the 2-D structured pretext layers to the inhibition head, while in GRIN light, we only connect the post model and the MoE output to the inhibition head. Interestingly, the light version performs well. So it may be effective to perform input selection from a global set of network activations.

**Gating mask location** $\mathcal{L}$. We investigate the optimal point of intervention for GRIN's inhibition signal by applying the gating mask at three distinct locations in the MoE architecture: (a) the mixture weights that determine expert contributions ($l^{\mathbb{IMW}}$), and (b) the combined output representation after expert aggre-

| Setting | Succ. rate (%) |
|---|---|
| Number of Recursion Steps | |
| 1 step (baseline) | $91.3 \pm 8.4$ |
| 2 steps | $76.7 \pm 2.5$ |
| 3 steps | $78.0 \pm 2.8$ |
| Number of Experts | |
| 3 Experts (baseline) | $91.3 \pm 8.4$ |
| 4 Experts | $78.7 \pm 0.9$ |
| 6 Experts | $82.0 \pm 4.3$ |
| Modulation and Gating Location | |
| $\mathbb{OR} + \mathbb{IMW}$ (baseline) | $91.3 \pm 8.4$ |
| $\mathbb{OR}$ only | $90.7 \pm 7.7$ |
| $\mathbb{IMW}$ only | $80.7 \pm 5.2$ |

Table 5: MT5 Ablation with GRIN

| Algorithm | 4 Experts | 6 Experts | 8 Experts |
|---|---|---|---|
| MOORE | $74.4 \pm 7.1$ | $71.5 \pm 9.3$ | $78.1 \pm 3.9$ |
| GRIN | $73.1 \pm 5.2$ | $78.3 \pm 6.5$ | $73.4 \pm 8.3$ |
| GRIN light | $76.3 \pm 5.4$ | $76.7 \pm 7.6$ | $76.6 \pm 5.4$ |

Table 6: Minigrid MT7 succ. rate (%)

---

[7]Experiments were run with 10 random seeds across 50 epochs.

gation ($l^{\text{OR}}$). We observe the inhibition mechanism is most effective at the output representation ($l^{\text{OR}}$) by selectively suppressing patterns in the aggregated output. The result aligns with the correlations in Table 4, and is helpful to inform optimal architectural integration of recurrent inhibition mechanisms.

# 5 GENERALIZING TO ML DOMAINS

The primary application of GRIN on MTRL shows its capability for state-space models. We make efforts to generalize GRIN to other ML domains. To ensure it is efficient to grow GRIN's parameter size, we explore a cascade of progressively increasing input diversity.

## 5.1 CASCADING INPUT DIVERSITY

Inhibitory neurons constitute a small portion of the neural population (Swanson & Maffei, 2019), and they exhibit remarkable diversity (Hofer et al., 2011; Pfeffer et al., 2013; Kajiwara et al., 2021). We focus on the diversity of input connections to translate this connection diversity to the GRIN head module ($\mathcal{G}$):

**Pretext Inhibition (PRE)**: The gating mechanism operates on incomplete forward-pass information, using the activations available up to the current layer (the pretext). Gated Linear Unit (GLU) is a special instance.

**Cross-iteration Global Inhibition (CIGI)**: We processes all activations from the previous stochastic optimization batch through a dedicated inhibition head to generate gating masks. Pooling operations are applied across the batch, producing sample-agnostic inhibition masks to gate the activations of the current iteration.

**Global Recurrent Inhibition Network (GRIN)**: GRIN algorithm was described in Section 3 which not only can produce specific masks per data sample, but also refine the adaptive gating decisions iteratively.

The inhibition head model combines information from multiple sources to compute the inhibition masks:

$$\mathbf{s}_{t+1} = \sigma(\mathcal{G}(\mathbf{a}_{global})) = \sigma(\mathbb{I}_{\text{PRE}} W_{\text{PRE}} \cdot f_{\text{PRE}}(\mathbf{a}_t) + \mathbb{I}_{\text{CIGI}} W_{\text{CIGI}} \cdot f_{\text{CIGI}}(\mathbf{a}_{t-1}^*) + \mathbb{I}_{\text{GRIN}} W_{\text{GRIN}} \cdot \mathbf{h}_t + b) \quad (3)$$

where $\mathbb{I}$ are indicator functions to select the existence of the connection; $\mathbf{a}_{t-1}^*$ represents pooled activations from the previous batch in CIGI. At training time for CIGI, the inhibition head and base model are jointly optimized. At inference time, when evaluating a new data point, we sample a batch of test data to simulate its prior iteration, producing the pooled inhibition masks.

| Algorithm | Acc. | Std. |
|---|---|---|
| Baseline model | 81.4% | 1.7e-2 |
| MoE 3/5 experts | 92.3% | 5.6e-2 |
| Random (dropout 0.25) | 94.9% | 2.0e-2 |
| Random (dropout 0.5) | 95.1% | 1.9e-2 |
| Random (dropout 0.75) | 95.9% | 8.1e-3 |
| One-layer I. (GLU) | 95.2% | 5.1e-3 |
| PRE | 95.3% | 8.4e-3 |
| CIGI | 94.9% | 2.3e-3 |
| GRIN | **96.3%** | 1.0e-2 |

Table 7: Results on mixed-numbers dataset

## 5.2 HAND-WRITTEN DIGITS AND NUMBER OF SQUARES

To generalize to the vision domain, we built a bi-modal simulated dataset comprising 120,000 samples, composed of half MNIST handwritten digits and half synthetic number patterns(Stoianov & Zorzi, 2012). Each sample is labeled with its numeric value (0-9). We use a convnet with a sparsely gated MoE with fully connected layers(Shazeer et al., 2017).

Results in Table 7 show that MoE models benefit from global inhibition gating. While dropout improves performance, it requires careful hyperparameter tuning. In contrast, adaptive inhibition methods automatically determine appropriate signal suppression levels, with performance scaling with connection diversity.

| Algorithm | 300K data | 1M data |
|---|---|---|
| Baseline model | 3.69e-6 | 1.17e-8 |
| MoE 3/10 experts | 3.68e-6 | 2.52e-10 |
| PRE | 3.37e-6 | 1.82e-10 |
| CIGI/GRIN | **3.31e-6** | **1.81e-10** |

Table 8: WMT English monolingual dataset (normalized log-likelihood loss)

## 5.3 LANGUAGE MODEL EXPERIMENT

We experiment with the transformer language model Vaswani et al. (2017); Shazeer et al. (2017); Du et al. (2022) with GRIN.

The MoE is applied on the fully connected layers of the trans-
former model, and gating heads is applied on the input of the MoE. The LM task is applied on WMT
English monolingual dataset (Maillard et al., 2024) with one smaller set containing 300K sentences
(7.5M words) and a larger one with 1 million sentences (25M words). Following the pre-processing,
we train transformers. Table 8 shows test set normalized log-likelihood on next word prediction.
The baseline MoE without gating performs similar to standard models, indicating less optimized
routing and gating. Global inhibition yields visible improvements across both datasets, confirming
its effectiveness for LM's.

## 6 CONCLUSION

We introduced the Global Recurrent Inhibition Network (GRIN), a novel architecture that enhances
dynamic routing in Mixture of Experts models. GRIN implements a global inhibition head that
recurrently processes diverse network activations and generate targeted gating signals. Our evalu-
ation demonstrates improvements across tasks, particularly MTRL. For future work, we study the
input spaces to GRIN with architecture specific designs and apply GRIN to more domains and in the
multi-modal space. These directions enable learned optimization of gating strategies that surpass
our current architectural design.

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

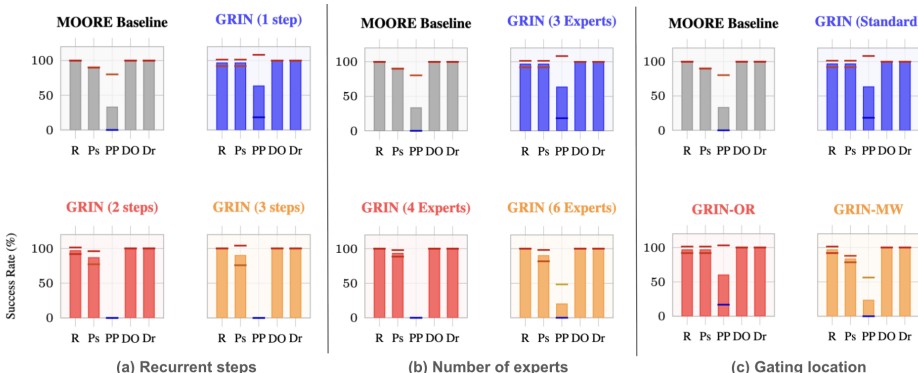

Figure 6: Ablation on MetaWorld MT5(a) Recurrent steps (b) Number of experts (c) Gating location. Task success rates (%) ± std bars for tasks R: Reach, Ps: Push, PP: Pick Place, DO: Door Open, Dr: Drawer

## A APPENDIX

### A.1 EXTENDED BACKGROUND AND LITERATURE REVIEW

Extensions to dropout and dynamic architectures include adapting parameters through additional network models in Hypernetworks (Ha et al., 2016) and HyperNEAT (Stanley et al., 2009). Neural architecture search uses reinforcement learning to choose network architectures (Pham et al., 2018; Zoph & Le, 2017; Zoph et al., 2018).

Related to Capsules, (Wang & Liu, 2018) offers an optimization perspective, (Kosiorek et al., 2019) proposes autoencoders with capsules, and (Rajasegaran et al., 2019) builds a deeper network.

Related to MTRL, multi-task learning exists in other domains such as supervised learning (Han et al., 2025) and recommender systems (Ma et al., 2018).

### A.2 MTRL GRIN IMPLEMENTATION DETAILS

We implement GRIN in pytorch using two system generalizations. First, we leverage the forward hook registered in pytorch *nn.Module* that triggers automated storing of any activations **a** and their gradient computation graph. This enables the storage of activations in efficient hash data-structures. Second the pytorch module traversal is used to search global and diverse sets of inputs for the inhibition head. For efficiency, we cache the activations before $l$ for the recurrent iterations, where $l$ is the earliest point of dynamic modulation and gating. The caching can be done for cases where inputs are non-sequential and identical.

### A.3 ABLATION BAR CHARTS ON METAWORLD MT5 (FIGURE. 7)

### A.4 MIXED-NUMBERS AND LANGUAGE MODEL DATA AND IMPLEMENTATION DETAILS

The data set is composed of 60,000 MNIST handwritten digits and 60,000 synthetic square number patterns following (Stoianov & Zorzi, 2012). The baseline architecture employs a two-layer convolutional network with max-pooling, augmented with a sparsely gated Mixture-of-Experts layer containing 5 experts (128-unit MLPs each). K=3 experts are selected per sample via a learned router. Results in Table 7 show that MoE models benefit from global inhibition gating for handling multi-modal inputs, models with more global inhibitory connections outperform those with local inhibition on this recognition task on multi-modal vision data.

Pre-processing details for the WMT dataset: we perform lower-casing, Porter stemming, digit replacement, contraction expansion, and punctuation removal. Transformer architecture: two-layers, embedding, hidden dimension is 50, 2 heads. We use vocabulary sizes of 8,500 and 15,000 for the

respective datasets (300k, 1m) and models are trained with Adam with a learning rate of 0.001 and batch size of 256. For each MoE we use 10 experts and each data points assigns K=3 experts.

## A.5 EXPERIMENTS AND ABLATION ON MINIGRID

Table 9: MiniGrid MT7 Results at Epoch 46: Average Return across 10 random seeds

| Algorithm | 4 Experts | 6 Experts | 8 Experts |
|---|---|---|---|
| MOORE | 0.7501 ± 0.0604 | **0.7746 ± 0.0606** | **0.7831 ± 0.0413** |
| GRIN | 0.7530 ± 0.0545 | 0.7523 ± 0.0490 | 0.7449 ± 0.0761 |
| GRIN light | **0.7759 ± 0.0612** | **0.7754 ± 0.0535** | 0.7657 ± 0.0543 |

Table 10: MiniGrid MT7 Performance Comparison at Epoch 46

| Experts | MOORE | GRIN | GRIN light |
|---|---|---|---|
| 4 | 0.7501 | 0.7530 | **0.7759** |
| 6 | 0.7746 | 0.7523 | **0.7754** |
| 8 | **0.7831** | 0.7449 | 0.7657 |

Table 11: MiniGrid MT7 Detailed Results at Epoch 46

| Experiment | Algorithm | Experts | Seeds | Average Return |
|---|---|---|---|---|
| GRIN with 4 experts | GRIN | 4 | 10 | 0.7530 ± 0.0545 |
| GRIN with 6 experts | GRIN | 6 | 10 | 0.7523 ± 0.0490 |
| GRIN with 8 experts | GRIN | 8 | 10 | 0.7449 ± 0.0761 |
| GRIN light with 4 experts | GRIN_light | 4 | 10 | 0.7759 ± 0.0612 |
| GRIN light with 6 experts | GRIN_light | 6 | 10 | 0.7754 ± 0.0535 |
| GRIN light with 8 experts | GRIN_light | 8 | 10 | 0.7657 ± 0.0543 |
| MOORE with 4 experts | MOORE | 4 | 10 | 0.7501 ± 0.0604 |
| MOORE with 6 experts | MOORE | 6 | 10 | 0.7746 ± 0.0606 |
| MOORE with 8 experts | MOORE | 8 | 10 | 0.7831 ± 0.0413 |
| **MOORE Average** | **MOORE** | **All** | - | **0.7693** |
| **GRIN Average** | **GRIN** | **All** | - | **0.7612** |

## A.6 DETAILED EXPERIMENT RESULTS

Table 12: MT50 Results Comparison at Epoch 4: MOORE Baseline (10 seeds) vs MOORE with GRIN (10 seeds)

| Task | Success Rate | | Mean J (reward) | | Discounted Mean J | |
|---|---|---|---|---|---|---|
| | Baseline | GRIN | Baseline | GRIN | Baseline | GRIN |
| **Overall Average** | **0.4650 ± 0.4769** | **0.4848 ± 0.4764** | - | - | - | - |
| Assembly | 0.0000 ± 0.0000 | 0.0000 ± 0.0000 | 231.23 ± 153.79 | 360.68 ± 185.50 | 109.23 ± 64.05 | 164.93 ± 77.42 |
| Basketball | 0.0000 ± 0.0000 | 0.0000 ± 0.0000 | 3.51 ± 0.74 | 5.40 ± 1.54 | 1.81 ± 0.36 | 2.69 ± 0.72 |
| Bin Picking | 0.0000 ± 0.0000 | 0.0000 ± 0.0000 | 3.88 ± 1.59 | 4.35 ± 1.60 | 1.98 ± 0.75 | 2.19 ± 0.74 |
| Box Close | 0.0000 ± 0.0000 | 0.0000 ± 0.0000 | 184.02 ± 21.20 | 195.41 ± 17.73 | 101.25 ± 10.36 | 106.43 ± 8.01 |
| Button Press Topdown | 0.8500 ± 0.3074 | 0.8600 ± 0.3007 | 794.87 ± 154.44 | 829.03 ± 116.02 | 323.64 ± 58.20 | 337.73 ± 42.97 |
| Button Press Topdown Wall | 0.7000 ± 0.4583 | 0.6600 ± 0.4363 | 728.81 ± 174.50 | 764.72 ± 122.01 | 298.81 ± 62.66 | 313.97 ± 45.32 |
| Button Press | 0.5400 ± 0.4055 | 0.6800 ± 0.4468 | 586.69 ± 267.03 | 666.91 ± 257.10 | 262.04 ± 112.79 | 295.88 ± 108.60 |
| Button Press Wall | 0.9500 ± 0.0671 | 0.8100 ± 0.3113 | 705.39 ± 143.22 | 703.72 ± 178.43 | 311.79 ± 59.57 | 308.64 ± 71.96 |
| Coffee Button | 0.9300 ± 0.1100 | 0.9700 ± 0.0900 | 856.20 ± 99.27 | 959.09 ± 80.96 | 392.75 ± 41.87 | 431.45 ± 35.74 |
| Coffee Pull | 0.0000 ± 0.0000 | 0.0000 ± 0.0000 | 7.30 ± 1.43 | 27.00 ± 51.27 | 3.74 ± 0.70 | 11.87 ± 20.81 |
| Coffee Push | 0.0000 ± 0.0000 | 0.1000 ± 0.3000 | 7.17 ± 0.98 | 77.27 ± 207.91 | 3.71 ± 0.42 | 29.56 ± 76.40 |
| Dial Turn | 0.7400 ± 0.1855 | 0.8600 ± 0.1114 | 802.69 ± 220.18 | 961.19 ± 103.92 | 358.42 ± 91.33 | 423.90 ± 47.50 |
| Disassemble | 0.0000 ± 0.0000 | 0.0000 ± 0.0000 | 59.09 ± 6.78 | 58.36 ± 1.64 | 31.18 ± 3.57 | 30.78 ± 0.87 |
| Door Close | 1.0000 ± 0.0000 | 1.0000 ± 0.0000 | 1029.50 ± 8.42 | 1025.88 ± 18.68 | 403.54 ± 5.19 | 401.60 ± 11.22 |
| Door Lock | 0.9800 ± 0.0600 | 0.9100 ± 0.1814 | 1000.94 ± 40.38 | 1009.83 ± 83.52 | 459.33 ± 22.30 | 467.73 ± 40.71 |
| Door Open | 0.3500 ± 0.4365 | 0.6700 ± 0.4428 | 631.00 ± 220.49 | 778.03 ± 189.37 | 272.43 ± 77.71 | 326.51 ± 64.03 |
| Door Unlock | 0.9800 ± 0.0600 | 0.9700 ± 0.0458 | 1163.95 ± 59.79 | 1180.55 ± 42.04 | 528.65 ± 28.07 | 537.89 ± 22.02 |
| Drawer Close | 1.0000 ± 0.0000 | 1.0000 ± 0.0000 | 1319.57 ± 71.68 | 1338.39 ± 28.96 | 625.54 ± 31.42 | 633.60 ± 12.69 |
| Drawer Open | 0.5800 ± 0.4750 | 0.8800 ± 0.2960 | 994.69 ± 192.38 | 1117.33 ± 112.02 | 467.22 ± 81.23 | 519.12 ± 47.54 |
| Faucet Open | 0.0000 ± 0.0000 | 0.0000 ± 0.0000 | 601.44 ± 8.63 | 610.43 ± 4.87 | 302.26 ± 4.04 | 306.25 ± 1.97 |
| Faucet Close | 0.8900 ± 0.2982 | 0.8900 ± 0.2982 | 1176.43 ± 189.00 | 1169.61 ± 204.14 | 537.53 ± 77.54 | 533.20 ± 83.86 |
| Hammer | 0.0900 ± 0.2119 | 0.1800 ± 0.3059 | 285.55 ± 211.38 | 507.40 ± 298.91 | 132.69 ± 86.29 | 225.14 ± 120.50 |
| Hand Insert | 0.2300 ± 0.2326 | 0.5300 ± 0.3926 | 162.31 ± 137.83 | 635.83 ± 367.14 | 64.69 ± 53.82 | 276.98 ± 160.43 |
| Handle Press Side | 1.0000 ± 0.0000 | 1.0000 ± 0.0000 | 1343.77 ± 12.74 | 1340.83 ± 5.25 | 634.97 ± 10.03 | 632.45 ± 4.03 |
| Handle Press | 1.0000 ± 0.0000 | 1.0000 ± 0.0000 | 1371.56 ± 15.18 | 1352.68 ± 27.42 | 659.50 ± 12.77 | 645.93 ± 17.97 |
| Handle Pull Side | 0.6000 ± 0.4899 | 0.4000 ± 0.4899 | 620.08 ± 506.85 | 404.53 ± 500.30 | 247.21 ± 203.04 | 167.70 ± 205.71 |
| Handle Pull | 0.3000 ± 0.4583 | 0.2300 ± 0.3951 | 660.42 ± 237.34 | 615.60 ± 213.67 | 295.31 ± 75.76 | 279.21 ± 70.74 |
| Lever Pull | 0.0000 ± 0.0000 | 0.0000 ± 0.0000 | 177.44 ± 6.72 | 174.44 ± 4.89 | 89.73 ± 2.79 | 88.60 ± 2.38 |
| Peg Insert Side | 0.0000 ± 0.0000 | 0.0000 ± 0.0000 | 4.61 ± 3.55 | 35.14 ± 92.28 | 2.24 ± 1.44 | 14.83 ± 38.07 |
| Peg Unplug Side | 0.0000 ± 0.0000 | 0.3100 ± 0.4346 | 5.24 ± 1.36 | 257.79 ± 387.04 | 2.77 ± 0.67 | 102.86 ± 153.54 |
| Pick Place Wall | 0.0000 ± 0.0000 | 0.0000 ± 0.0000 | 0.00 ± 0.00 | 0.00 ± 0.00 | 0.00 ± 0.00 | 0.00 ± 0.00 |
| Pick Out Of Hole | 0.0000 ± 0.0000 | 0.0000 ± 0.0000 | 2.68 ± 0.43 | 2.86 ± 0.22 | 1.38 ± 0.22 | 1.48 ± 0.11 |
| Pick Place | 0.0000 ± 0.0000 | 0.0000 ± 0.0000 | 2.47 ± 0.78 | 3.65 ± 0.86 | 1.28 ± 0.39 | 1.89 ± 0.43 |
| Plate Slide | 0.9800 ± 0.0400 | 0.9300 ± 0.1418 | 1075.83 ± 83.20 | 1044.71 ± 83.52 | 455.12 ± 38.62 | 445.87 ± 32.55 |
| Plate Slide Side | 0.9200 ± 0.1470 | 0.9100 ± 0.1814 | 914.24 ± 132.92 | 1009.38 ± 128.67 | 422.18 ± 53.74 | 457.21 ± 40.24 |
| Plate Slide Back | 0.9900 ± 0.0300 | 1.0000 ± 0.0000 | 1240.71 ± 47.59 | 1258.33 ± 50.87 | 562.90 ± 28.33 | 574.76 ± 22.92 |
| Plate Slide Back Side | 1.0000 ± 0.0000 | 0.8100 ± 0.2809 | 1252.73 ± 14.69 | 1127.90 ± 222.28 | 567.86 ± 9.06 | 510.63 ± 97.01 |
| Push Back | 0.0100 ± 0.0300 | 0.0100 ± 0.0300 | 7.88 ± 2.29 | 11.29 ± 6.97 | 3.79 ± 1.10 | 5.29 ± 3.00 |
| Push | 0.2450 ± 0.2307 | 0.1850 ± 0.2098 | 299.74 ± 244.55 | 329.69 ± 212.78 | 133.77 ± 108.93 | 150.47 ± 95.01 |
| Push Wall | 0.0100 ± 0.0300 | 0.0000 ± 0.0000 | 185.91 ± 266.21 | 363.38 ± 258.71 | 85.37 ± 116.02 | 168.76 ± 115.87 |
| Reach | 0.9400 ± 0.0663 | 0.9600 ± 0.0490 | 1342.59 ± 12.75 | 1346.45 ± 7.40 | 641.34 ± 7.54 | 642.01 ± 4.92 |
| Reach Wall | 0.9700 ± 0.0458 | 0.9600 ± 0.1200 | 1318.39 ± 14.23 | 1315.76 ± 30.08 | 622.43 ± 5.93 | 621.07 ± 14.27 |
| Shelf Place | 0.0000 ± 0.0000 | 0.0000 ± 0.0000 | 0.01 ± 0.03 | 0.06 ± 0.16 | 0.01 ± 0.02 | 0.03 ± 0.07 |
| Soccer | 0.4300 ± 0.3848 | 0.4600 ± 0.3262 | 427.37 ± 363.84 | 419.29 ± 252.71 | 180.68 ± 153.76 | 180.90 ± 105.18 |
| Stick Pull | 0.0000 ± 0.0000 | 0.0000 ± 0.0000 | 4.75 ± 1.59 | 5.31 ± 0.83 | 2.46 ± 0.75 | 2.75 ± 0.40 |
| Sweep Into | 0.5800 ± 0.4771 | 0.7300 ± 0.4051 | 785.79 ± 387.01 | 1001.91 ± 305.98 | 358.69 ± 169.19 | 457.60 ± 134.81 |
| Sweep | 0.0000 ± 0.0000 | 0.0000 ± 0.0000 | 168.00 ± 72.12 | 256.93 ± 171.44 | 79.93 ± 33.35 | 117.49 ± 68.80 |
| Window Open | 1.0000 ± 0.0000 | 0.9900 ± 0.0300 | 1016.99 ± 41.21 | 1022.67 ± 50.52 | 410.08 ± 20.59 | 411.92 ± 24.61 |
| Window Close | 1.0000 ± 0.0000 | 0.9000 ± 0.3000 | 1045.53 ± 79.82 | 929.26 ± 269.54 | 427.96 ± 40.67 | 379.61 ± 106.33 |

Table 13: MT3 Results (3 Tasks) - Mean ± Std across 5 seeds

| Experiment | Task | Success Rate | Mean J (Reward) | Discounted Mean J |
|---|---|---|---|---|
| MT3 Baseline (256hd, 2 layers) | Reach-v2 | 1.000 ± 0.000 | 1336.18 ± 16.97 | 631.05 ± 13.89 |
| | Push-v2 | 0.100 ± 0.200 | 45.86 ± 60.89 | 26.69 ± 35.61 |
| | Pick-Place-v2 | 0.000 ± 0.000 | **2.15 ± 1.43** | **1.17 ± 0.74** |
| MT3 Pretext Inhi. (weights only, 256hd, 2 layers) | Reach-v2 | **1.000 ± 0.000** | **1346.20 ± 14.14** | **642.27 ± 10.18** |
| | Push-v2 | **0.100 ± 0.200** | **62.23 ± 55.93** | **29.98 ± 23.95** |
| | Pick-Place-v2 | 0.000 ± 0.000 | 1.67 ± 0.36 | 0.96 ± 0.19 |

Table 14: MT5 Results (5 Tasks) - Mean ± Std across 5 seeds

| Experiment | Task | Success Rate | Mean J (Reward) | Discounted Mean J |
|---|---|---|---|---|
| MT5 Baseline (256hd, 2 layers) | Reach-v2 | 1.000 ± 0.000 | 1350.54 ± 15.72 | 644.56 ± 12.72 |
| | Push-v2 | 0.000 ± 0.000 | 33.10 ± 25.92 | 20.36 ± 16.97 |
| | Pick-Place-v2 | 0.000 ± 0.000 | 1.85 ± 0.44 | 1.06 ± 0.28 |
| | Door-Open-v2 | 0.000 ± 0.000 | 458.59 ± 44.02 | 215.29 ± 21.55 |
| | Drawer-Open-v2 | 0.000 ± 0.000 | **738.09 ± 25.11** | **358.31 ± 9.54** |
| MT5 Pretext Inhibition (weights + features, 256hd, 2 layers) | Reach-v2 | 1.000 ± 0.000 | 1340.13 ± 15.30 | 636.03 ± 11.92 |
| | Push-v2 | 0.000 ± 0.000 | 17.82 ± 17.18 | 8.86 ± 7.57 |
| | Pick-Place-v2 | 0.000 ± 0.000 | 1.46 ± 0.49 | 0.87 ± 0.28 |
| | Door-Open-v2 | 0.000 ± 0.000 | 439.82 ± 120.14 | 203.16 ± 47.28 |
| | Drawer-Open-v2 | 0.000 ± 0.000 | 675.73 ± 110.36 | 331.72 ± 48.61 |
| MT5 Pretext Inhibition (weights only, 256hd, 2 layers) | Reach-v2 | **1.000 ± 0.000** | **1351.29 ± 16.27** | **648.46 ± 7.86** |
| | Push-v2 | 0.000 ± 0.000 | 10.97 ± 8.08 | 6.31 ± 5.08 |
| | Pick-Place-v2 | 0.000 ± 0.000 | 1.68 ± 0.38 | 0.94 ± 0.33 |
| | Door-Open-v2 | 0.000 ± 0.000 | 457.90 ± 85.74 | 213.64 ± 34.67 |
| | Drawer-Open-v2 | 0.000 ± 0.000 | 716.92 ± 30.14 | 349.03 ± 14.12 |
| MT5 GRIN (rec=1) (weights + features, 256hd, 2 layers) | Reach-v2 | 1.000 ± 0.000 | 1311.13 ± 50.76 | 624.66 ± 19.35 |
| | Push-v2 | 0.000 ± 0.000 | **43.07 ± 18.62** | **21.90 ± 10.27** |
| | Pick-Place-v2 | 0.000 ± 0.000 | 2.02 ± 1.07 | 1.18 ± 0.59 |
| | Door-Open-v2 | 0.000 ± 0.000 | 459.95 ± 33.94 | 217.50 ± 12.51 |
| | Drawer-Open-v2 | 0.000 ± 0.000 | 718.56 ± 21.37 | 350.56 ± 8.39 |
| MT5 GRIN (rec=3) (weights + features, 256hd, 2 layers) | Reach-v2 | 0.900 ± 0.200 | 1306.11 ± 72.21 | 627.65 ± 28.29 |
| | Push-v2 | 0.000 ± 0.000 | 36.23 ± 22.20 | 18.57 ± 10.37 |
| | Pick-Place-v2 | 0.000 ± 0.000 | **2.37 ± 1.56** | **1.26 ± 0.80** |
| | Door-Open-v2 | 0.000 ± 0.000 | 409.93 ± 91.58 | 195.16 ± 40.73 |
| | Drawer-Open-v2 | 0.000 ± 0.000 | 733.90 ± 48.66 | 356.60 ± 23.27 |
| MT5 GRIN (rec=1) (weights only, 256hd, 2 layers) | Reach-v2 | 0.900 ± 0.200 | 1313.09 ± 58.16 | 626.27 ± 24.21 |
| | Push-v2 | 0.000 ± 0.000 | 26.70 ± 7.30 | 15.13 ± 3.82 |
| | Pick-Place-v2 | 0.000 ± 0.000 | 1.95 ± 0.59 | 1.06 ± 0.23 |
| | Door-Open-v2 | 0.000 ± 0.000 | **461.62 ± 17.06** | **217.69 ± 7.55** |
| | Drawer-Open-v2 | 0.000 ± 0.000 | 722.52 ± 17.35 | 351.05 ± 9.82 |

Table 15: MT10 Results Comparison at Epoch 20: MOORE Baseline (9 seeds) vs MOORE with GRIN (10 seeds)

| Task | Success Rate | | Mean J | | Discounted Mean J | |
|---|---|---|---|---|---|---|
| | Baseline | GRIN | Baseline | GRIN | Baseline | GRIN |
| **Overall Average** | **0.8844 ± 0.3137** | **0.8820 ± 0.3135** | - | - | - | - |
| Reach | 0.9778 ± 0.0629 | 0.9700 ± 0.0900 | 1350.86 ± 9.58 | 1351.25 ± 12.89 | 643.48 ± 7.06 | 645.07 ± 9.72 |
| Push | 1.0000 ± 0.0000 | 0.9500 ± 0.1025 | 1165.10 ± 92.92 | 1180.19 ± 94.50 | 507.36 ± 61.12 | 526.68 ± 53.01 |
| Pick Place | 0.0000 ± 0.0000 | 0.0000 ± 0.0000 | 2.78 ± 0.84 | 4.83 ± 2.01 | 1.42 ± 0.43 | 2.45 ± 0.92 |
| Door Open | 1.0000 ± 0.0000 | 1.0000 ± 0.0000 | 1067.50 ± 21.55 | 1043.79 ± 45.21 | 445.38 ± 12.66 | 432.47 ± 21.60 |
| Drawer Open | 0.8778 ± 0.3119 | 1.0000 ± 0.0000 | 1203.17 ± 165.46 | 1266.55 ± 22.76 | 562.34 ± 71.38 | 587.66 ± 11.94 |
| Drawer Close | 1.0000 ± 0.0000 | 1.0000 ± 0.0000 | 1359.86 ± 4.67 | 1354.25 ± 4.49 | 647.33 ± 4.10 | 642.47 ± 3.87 |
| Button Press Topdown | 1.0000 ± 0.0000 | 1.0000 ± 0.0000 | 906.88 ± 14.24 | 880.70 ± 25.45 | 372.22 ± 7.41 | 358.76 ± 12.05 |
| Peg Insert Side | 0.9889 ± 0.0314 | 0.9000 ± 0.3000 | 1096.05 ± 30.69 | 1037.75 ± 144.47 | 453.41 ± 13.36 | 423.62 ± 59.19 |
| Window Open | 1.0000 ± 0.0000 | 1.0000 ± 0.0000 | 1083.37 ± 17.57 | 1078.88 ± 22.52 | 443.51 ± 11.29 | 446.55 ± 13.32 |
| Window Close | 1.0000 ± 0.0000 | 1.0000 ± 0.0000 | 1063.78 ± 45.47 | 1082.34 ± 21.52 | 435.26 ± 23.63 | 444.50 ± 13.98 |

