# OpenReview forum: "Global Neural Inhibition Improves Mixture of Experts in Multi-task Reinforcement Learning"
_ICLR.cc/2026/Conference — ICLR 2026 Conference Withdrawn Submission_

### Official Review · Reviewer_eCJY · 2025-10-17

**Soundness:** 1
**Presentation:** 2
**Contribution:** 1
**Rating:** 2
**Confidence:** 5

**Summary:**

The authors propose an extension of MoEs by which they modulate features to route data more effectively through the network. They validate this approach through experiments on MT10 and MT50 from Meta-World, alongside experiments in vision and language modelling.

**Strengths:**

Effective architecture design that can be applied across multiple problem settings is a useful idea which should be more widely explored.

**Weaknesses:**

1. The largest weakness are the MTRL results. Recent work has found that the results produced by the PaCo and MOORE papers was inaccurate [1]. Due to the re-using of the results from MOORE, I ask the authors to run the relevant baselines under the same benchmark settings (reward version, specifically) to enable effective comparison between results. From my understanding, [1] produced a set of benchmark algorithms that should be easily run.

2. The need of specialized architectures, especially in homogeneous MTRL (i.e. tasks share same action & state spaces), has been found to be inaccurate while the driver of performance is in fact the parameter count [2]. The results of MOORE, for example, can be matched/outperformed by a similarly scaled feed-forward neural network as found in [2]. How does the proposed method compare to the findings in [2], that suggests that specialized architectures are not needed in MTRL?

3. The authors need to update their results to use a measure such as IQM [3] to ensure that their results are statistically significant. The current mean & std formulation shows large overlaps in performance which clouds statistical significance.

4. This is a common misconception, but there is no such thing as 'mt10-rand' or 'mt10-fixed.' Early works used Meta-World incorrectly and introduced this odd nomenclature. Due to it's goal conditioned nature, Meta-World should only be 'mt10-rand' (but you don't need to refer to it as such) where new goals are sampled upon resetting.


Citations
1. https://arxiv.org/abs/2505.11289

2. https://arxiv.org/abs/2503.05126

3. Rishabh Agarwal, Max Schwarzer, Pablo Samuel Castro, Aaron Courville, Marc G. Bellemare (2021). Deep RL at the Edge of the Statistical Precipice. NeurIPS

**Questions:**

How could the method be extended to be multi-modal? Could the method be extended to the heterogeneous MTRL setting?

---

> ### Author Response · Authors · 2025-12-01
>
> We thank reviewer eCJY for your insightful, effective, and impactful comments!
>
> "The largest weakness is the MTRL results. Recent work has found that the results produced by the PaCo and MOORE papers was inaccurate [1]....[1] produced a set of benchmark algorithms that should be easily run."
>
> - In fact, we took the reviewer’s suggestions and got the JAX code running right away. We alap implemented a version of GRIN. We found that the MOORE MT50 scripts give about a 67% success rate, and it'd be what we use for experiments going forward. We were also able to implement different versions of GRIN with sequences, but we decided there is insufficient time produce full MT50 results for GRIN during rebuttal. We thank you for this important pointer and will continue with Metaworld+. With the resources and time we have, we report all comprehensive experiments performed with the MOORE code as baselines and our GRIN implementations.
>
>
> "The need for specialized architectures, especially in homogeneous MTRL (i.e., tasks share the same action & state spaces), has been found to be inaccurate while the driver of performance is in fact the parameter count [2]. The results of MOORE, for example, can be matched/outperformed by a similarly scaled feed-forward neural network as found in [2]. How does the proposed method compare to the findings in [2], which suggest that specialized architectures are not needed in MTRL?"
>
> - With the JAX code above, we also performed experiments (XP-A) with forward networks with hidden dimensions 2048, which arrived at similar results as MOORE, ~68-69% success rate. So it's perhaps less obvious to us whether the parameter scaling reported in [2] is effective without us replicating the results. Further, we also tried MOORE with 600 hidden units (XP-B), and the MT50 results are also around 68%. From our perspective, we could improve results by a significant +3.7% with GRIN, which adds about the same number of additional parameters as (XP-A) and (XP-B), but reaches greater improvement than (XP-A) and (XP-B). Further, it's our plea to refer to the extensive ablation studies and detailed experiments in this paper, with supervised learning in vision, sequence language models, which reached the conclusion that gating/modulation helps and improves results.
>
> "The authors need to update their results to use a measure such as IQM [3] to ensure that their results are statistically significant. The current mean & std formulation shows large overlaps in performance which clouds statistical significance."
>
> - Thank you for this comment, and we've updated both MT10 and MT50 with IQM metrics! It does help a lot and the results are statistically significant. Really appreciate this constructive comment and help, kindly refer to the revised paper and glad to improve further.
>
> "This is a common misconception, but there is no such thing as 'mt10-rand' or 'mt10-fixed.' Early works used Meta-World incorrectly and introduced this odd nomenclature. Due to it's goal conditioned nature, Meta-World should only be 'mt10-rand' (but you don't need to refer to it as such) where new goals are sampled upon resetting."
> - Thank you for pointing this out! We agree with your comment, and we've revised the paper accordingly in the experiment setup and sections. Much appreciate it and let us know any further suggestions here.
>
> Questions:
> "How could the method be extended to be multi-modal? Could the method be extended to the heterogeneous MTRL setting?"
> - Definitely, we could make the method multi-modal, such as in a VQA setting - using visual signals to modulate potential text outputs. In the 'generalizing to other domains' section, we started proving this, and it shows potential to work across domains. This paper is mostly about RL and MTRL though, so we'll focus on that and plan future efforts on multi-modal.
> - Thanks for the comment. My thinking is that dynamic gating will work better where modulation or decision is needed on parameter choice or composition. In a heterogeneous MTRL setting, there will be more need to deal with the diversity of models, data, tasks, perhaps rewards, so modulation will make a more visible impact.

---

### Official Review · Reviewer_HBZF · 2025-10-27

**Soundness:** 1
**Presentation:** 2
**Contribution:** 1
**Rating:** 2
**Confidence:** 4

**Summary:**

The paper proposes GRIN (Global Recurrent Inhibition Networks), a mechanism for dynamically modulating mixture-of-experts (MoE) architectures using global network activations. The approach introduces an "inhibition head" that observes activation patterns across the network and generates sigmoid gating masks applied at three locations: input representations (IR), mixture weights (MW), and output representations (OR). The paper presents a recurrent formulation that iteratively refines these masks over T timesteps using backpropagation through time. Empirical evaluation focuses on MetaWorld multi-task reinforcement learning benchmarks (MT10 and MT50), with additional demonstrations on a synthetic vision dataset and limited-scale language modeling.

**Strengths:**

**Clear and Modular Architectural Design.** The paper proposes a modular inhibition head that can be integrated with existing MoE architectures, targeting three distinct modulation points (IR, MW, OR).

**Competitive MT10 Performance.** The reported 89.4 ± 1.0% success rate on MetaWorld MT10 at 20M steps represents good empirical performance.

**Cross-Domain Evaluation Attempt.** The paper evaluates on RL, vision, and language tasks, demonstrating an effort to show generality beyond a single domain. This attempt to demonstrate broader applicability is valuable even if the non-RL experiments are limited in scale.

**Weaknesses:**

**1. Central Contradiction: Recurrence Motivation Not Supported by Evidence.**

The paper's title (*pdf title mentions "recurrent", openreview title omits it*), abstract, and introduction strongly emphasize "Recurrent Inhibition Networks" as the core contribution. Lines 42-44 explicitly motivate recurrence by stating that feedforward architectures "*make routing decisions based on current input alone, without information from later layers.*" The proposed solution is described as a recurrent formulation that "*iteratively refines modulation decisions*" (line 45).

However, Table 3 directly contradicts this central claim through the ablation on recurrence depth:
- 1 step (T=1): 91.3 ± 8.4%
- 2 steps (T=2): 76.7 (15 percentage point drop)
- 3 steps (T=3): 78.0 (13 percentage point drop)

These performance degradations are substantial and fall well outside the baseline's error bars when leveraging the central mechanism the paper proposes. The paper kind of tacitly acknowledges this with the brief explanation in L311 "a single recurrence step is typically sufficient for less complex MTRL tasks" (line 311). So in essence it means that without recurrence is how we get best results?

- **At T=1, there is no recurrence.** The system performs a single forward pass where an RNN module processes one timestep. The recurrent connections, the defining feature of RNNs that enables integration over multiple steps, are never utilized. What remains is effectively a feedforward network with a particular parameterization.

- **Almost all main results use T=1**. The paper explicitly uses T=1 for MT50 (line 266, 'recurrence depth of 1, as determined optimal in our ablation studies'). For MT10, while the recurrence depth is not explicitly stated, it would be inconsistent to use T>1 given that: (a) the ablation shows T=1 is optimal and T>1 severely degrades performance, and (b) MT50 explicitly uses T=1 based on these ablations. If the paper did use T>1 for MT10, this would contradict their own ablation findings and should be clarified.

- **The paper never demonstrates where T>1 helps.** The explanation references "less complex" tasks, yet MT50 (50 tasks, substantially more complex) also uses T=1. No experiments test whether harder tasks benefit from multiple recurrence steps.

- **The recurrence formulation is both unused *and* conceptually flawed.** On top of the paper not leveraging recurrence (T>1) in any main results the formulation itself is mismatched with its motivation. The paper motivates recurrence as accessing "information from later layers" (lines 42-44), yet states "*we use the same x to propagate multiple time-steps*" (line 169). With repeated processing of the same input, the system at best observes how layers responded in previous iterations, not how downstream layers respond to current gating decisions in the architectural sense. The paper never specifies whether $a_{global}$​ includes later-layer activations.

Overall, all of these seem to represent a fundamental misalignment between what the paper claims to contribute (recurrent inhibition that accesses later-layer information) and what actually works (global feedforward inhibition at T=1).


**2. Mathematical specification insufficient for reproduction**
- **Undefined $a_{\text{global}}$.** The paper never specifies which layers feed $a_{\text{global}}$, how heterogeneous tensors are pooled/aggregated, or the resulting shape.

- **Mask dimension mismatch.** Eq. 1 sets $s^{l_{\text{IR}}}, s^{l_{\text{MW}}}, s^{l_{\text{OR}}}=\sigma(G(a_{\text{global}}))$, but these masks require *different sizes* ($\mathbb{R}^d$ for IR/OR, $\mathbb{R}^N$ for MW). No multi-head decoder, per-site projections, or broadcasting rules are described.

- **Objective over steps unclear.** Text suggests per-step losses are summed ($L=\sum_t L_t$), which scales gradients with $T$; if only $L_T$ is used, it isn’t stated.

In general key specs are missing, so correctness can’t be verified and reproduction requires guesswork.

**Further issues**
- **MT50 incomplete & MT10 not statistically validated.** MT50 is reported only at 10M/16M (one case **1 seed**), although authors mention preliminary so the results cant be considered with much weight. The MT10 SOTA gap ($89.4 \pm 1.0$ vs. $88.4 \pm 3.4$) lacks is too narrow to claim SOTA imo without more rigorous statistical validations.

- **CV/LM results are marginal and under-contextualized.** Vision shows a ~0.6 pp gain with overlapping variance; LM reports losses on unusual $10^{-6}$–$10^{-10}$ scales without calibration.


- **Ablation inconsistencies & OPE mislabeling.**
- Table 3 favors **3 experts** yet main runs use **4/6** without rationale; OR-only ≈ OR+MW (MW-only lags), implying gains come mainly from output modulation.
- Section 4.3’s “FQE” seems to be Gaussian Q-fit/correlation instead, can the authors comment on this?

**Questions:**

1. **Recurrence:** Since $T>1$ degrades performance and main results use $T=1$, where (if anywhere) does recurrence help, and by what mechanism?
2. Precisely define $a_{\text{global}}$ (layers and aggregation), explain how $G$ produces IR/MW/OR masks of different sizes (e.g., separate heads), and state the objective over steps ($L_T$ vs $\sum_t L_t$) and any normalization/clipping.
3. **CIGI:** How is deterministic single-sample inference performed given dependence on prior-batch activations, and what do ablations show about CIGI’s contribution?
4. **Design choices:** Why use 4/6 experts when ablations favor 3, and is MW gating necessary given OR-only $\approx$ OR+MW?

---

> ### Author Response · Authors · 2025-12-03
>
> We thank reviewer HBZF for your insightful and detailed comments!
>
> Many thanks for your kind review and words on the paper structure, MT10 results, and the cross-domain efforts. We appreciate that you found time to help read and understand our paper and the insightful comments.
>
> About the T=1 and recursion feedback, we acknowledge the limitation that the current implementation with MOORE results in better results at T=1. This was not our intention, but rather it's an empirical result. We believe the limitation arises due to the limited temporal context in our current implementation, and the implementation is limited by the complexity of SAC with MoE plus G-S process. As far as what we can present for this submission, we will retain the results using MOORE implementation to help the community align and improve the Metaworld benchmark, while we already implemented and are testing a fully sequential version of GRIN with sequential data input. We won't be able to produce convincing results due to the Metaworld MT50 alignment per the overall statement; however, we look forward to presenting that in the next version of the paper.
>
> "Undefined $a_{\text{global}}$. The paper ..."
> - We are adding this detail to the appendix, while it includes most of the activations in the actor or critic networks, respectively.
>
> "Mask dimension mismatch. Eq. 1 sets $s^{l_{\text{IR}}}, s^{l_{\text{MW}}}, s^{l_{\text{OR}}}=\sigma(G(a_{\text{global}}))$, but these masks require different sizes... "
> - We understand this may be confusing without explicit model definition in the appendix, we are glad to add that. While the multi-head (OR, IR, and MW each have a head that projects into the respective dimensions) architecture is able to customize dimensions as needed. Thank you for discussing this presentation issue.
> - We understand the confusion comes from $s^{l_{\text{IR}}}, s^{l_{\text{MW}}}, s^{l_{\text{OR}}}=\sigma(G(a_{\text{global}}))$. If we present it as $h_g = H(a_{\text{global}}), \quad s^{l_{\text{IR}}} = \sigma h_{l_{\text{IR}}}(h_g), \quad s^{l_{\text{MW}}} = \sigma h_{l_{\text{MW}}}(h_g), \quad s^{l_{\text{OR}}} = \sigma h_{l_{\text{OR}}}(h_g)$, perhaps this is more clear. Each equation explains the distinct head that projects to each of the masking locations.
>
> "Objective over steps unclear. Text suggests per-step losses are summed ($L=\sum_t L_t$), which scales gradients with $T$; if only $L_T$ is used, it isn’t stated."
> - Thank you for raising this issue, and we understand it may be a presentation with an unclear description. While we believe the text is quite clear saying the per-step losses are summed: ($L=\sum_t L_t$). We can add a sentence that says we did not use only $L_T$ (the last step loss) as the loss function. Thank you.
>
> "In general key specs are missing, so correctness can’t be verified and reproduction requires guesswork."
> - We'd be glad to share more key specs in the appendix. Thank you.
>
> "MT50 incomplete & MT10 not statistically validated..."
> - Please kindly see the revised version where the IQM is used and the data is now validated with statistical significance. Thank you.
>
> "CV/LM results are marginal and under-contextualized. ... "
> - Thank you, and we show small-scale results which are illustrative of how the method could generalize to other domains. We hope not to claim on vision results or text results in this paper. The smallness in gains is less of a concern, while we actually hope to present the fact that 'diversity' or 'globalness' of connections improves results, even if the improvements are small. This offers technical validation and intuition towards showing GRIN could be applied in neural networks in general. While we understand your concern, and have helped improve how we present the results in the last section.
>
> "Table 3 favors 3 experts yet main runs use 4/6 without rationale; OR-only ≈ OR+MW (MW-only lags), implying gains come mainly from output modulation."
> - Thank you for the insightful observation. This fact: "gains come mainly from output modulation" actually supports the main claims of the paper - we claim that GRIN is effective in dynamic gating and modulation with sigmoid gates, as opposed to changing mixture weights and routing for the MoE. This is an empirical finding, and with more related papers coming out, this also aligns with the literature (see the new paper quoted in the global comment).
>
> "Section 4.3’s “FQE” seems to be a Gaussian Q-fit/correlation instead, can the authors comment on this?"
> - FQE is Fitted Q-Evaluation, and a commonly used probability distribution is Gaussian. Since we are fitting to the Q values, the results are shown empirically under the assumption that the data is Gaussian. The difference in the mean of the fitted Gaussians indicates the improvement in value when we adopt the model-recommended policy.

---

### Official Review · Reviewer_5irm · 2025-10-31

**Soundness:** 2
**Presentation:** 2
**Contribution:** 3
**Rating:** 2
**Confidence:** 3

**Summary:**

This paper introduces Global Recurrent Inhibition Networks (GRIN), a biologically inspired mechanism for dynamic modulation in Mixture-of-Experts (MoE) architectures. GRIN adds a global inhibitory head that aggregates activations from across the network and produces gating masks applied at multiple points (input representation, mixture weights, and output representation). The inhibition is recurrent, iteratively refining the modulation over several time steps, akin to LSTM-style gating.

The authors evaluate GRIN primarily in multi-task reinforcement learning (MTRL) using MetaWorld (MT10, MT50), achieving strong results compared to prior MoE-based MTRL methods (PaCo, MOORE). They also test GRIN on a small vision dataset (mixed MNIST/squares) and a language modeling task (WMT-En) to show generalization. Ablations investigate recurrence depth, gating locations, number of experts, and the impact of inhibition masks on Q-values. The results suggest that global inhibitory modulation can stabilize and improve multi-task learning dynamics.

**Strengths:**

- The idea of using recurrent global inhibition for adaptive gating in MoE architectures is interesting. The paper connects biological inhibition mechanisms with modern sparse neural architectures in a plausible way.
- The recurrent formulation (Eq. 2) and the integration of inhibition masks at different architectural locations are coherent. It could be attached to existing MoE architectures with minimal changes.
- The use of MetaWorld and ablations provides credible empirical grounding.
- Improving stability and specialization in MTRL MoEs is relevant. GRIN could inspire new designs for adaptive routing or modulation in large-scale RL and multi-domain systems.
- GRIN achieves consistent improvements on MT10 and MT50 over strong baselines (PaCo, MOORE).

**Weaknesses:**

- The paper lacks a solid theoretical explanation or variance analysis for why recurrent inhibition improves stability. The analogy to biological inhibition is interesting but remains qualitative.
- The work’s main claim of “general applicability” is under-supported. The vision and LM results are limited to small synthetic or low-scale datasets, leaving unclear whether GRIN scales to large-scale LLMs or complex visual tasks.
- Although several ablations are presented, they omit key comparisons such as ablations without recurrence but with global inputs, or with local recurrent gating only. This makes it difficult to isolate which component drives the gains.
- Confidence intervals and significance tests are missing for some results (especially MT50). Some tables include partial seeds or intermediate checkpoints, which weakens reproducibility.
- Some design and training details are underspecified, for example, hidden dimension of the inhibition head, update frequency of global activations, and precise backprop-through-time length. This may limit re-implementation.
- The relation to prior MoE gating approaches (e.g., Expert Choice, Switch Transformers, or ST-MoE) is discussed only superficially. It is unclear whether GRIN could integrate or conflict with these systems.
- The exposition is dense and occasionally ambiguous. Some notation and definitions are introduced without sufficient explanation, making sections hard to follow. Figures and tables could be more self-contained and clearly labeled, as captions often lack essential context. Overall, the low clarity and incomplete descriptions reduce the paper’s accessibility and reproducibility.

**Questions:**

- Can you provide variance reduction or gradient-norm analyses to support the claim that GRIN stabilizes optimization compared to local gating or MOORE?
- What is the computational overhead (in FLOPs or wall-time) introduced by the recurrent inhibition step?
- Did you test GRIN in single-task RL or standard supervised settings to check whether the inhibition mechanism harms specialization when no task diversity is present?
- How sensitive is GRIN to the number of recurrence steps T and the update frequency of global activations?
- Could GRIN be combined with routing regularizers (like load-balancing or entropy penalties) used in large-scale MoEs?
In the language modeling results, what is the model scale (parameter count) and training compute? Are improvements statistically significant across seeds?
- What is the interpretation of negative correlations between mixture-weight inhibition and Q-value improvement (Table 4)?
- How is backpropagation-through-time truncated or regularized to avoid exploding gradients during recurrent inhibition?
- How does the recurrent inhibition affect gradient flow and credit assignment across experts? Have you visualized gradient magnitudes through time?
- Is the recurrent head shared across layers or instantiated per MoE layer? Clarify parameter sharing and scaling implications.
- I was wondering how does GRIN behave as the number of experts N grows (e.g., 8→64)? Is the global inhibition signal still efficient?
- Have you tested inference latency when the recurrent inhibition loop is unrolled?

**Details Of Ethics Concerns:**

There are no ethics concerns to report.

---

> ### Author Response · Authors · 2025-12-01
>
> We thank reviewer 5irm for the in-depth, insightful, and helpful comments.
>
> "The paper lacks a solid theoretical explanation or variance analysis... "
> - We would appreciate a theoretical and mathematical analysis of the algorithm. In the next revision, we plan to add the error and stability analysis, just like the classical LSTM paper in 1997: https://www.bioinf.jku.at/publications/older/2604.pdf  I supoose we were a bit wrought with the MT50 dataset.
>
> "The work’s main claim of general applicability is under-supported..."
> - Thank you for this insightful comment. In this rebuttal revision, we realize this limitation and have carefully revised the introduction narrative. We'd limit the claim to a global view of the network, and carry forward temporal context to improve gating. It's our view this gating approach will be impactful in the complex dynamical systems of MTRL. We hope it can be a technique that could be adopted by RL researchers.
>
> "Although several ablations are presented, they omit key comparisons such as ablations..."
> - We trust the T=1 case may cover the case with global activations. In fact, the question is that T must be >=1 if we are to be able to take in later layer activations such as Q. Due to our resource limitations, as described in the overall comment, it was challenging to perform ablation on MT50.
>
> "Confidence intervals and significance tests are missing for some results (especially MT50)..."
> - Thank you for your comment and we trust that the revised version have full 100M results and with IQM results together with bounds. Feel free to let us know if it could be further improved.
>
> "Some design and training details are underspecified, for example, hidden dimension..."
> - Thank you, and these details are added or being added to the revised version. The hidden dimensions in the GRIN module (before heads) are 2 x the common hidden dimension in the critic/actor network (400), so it's 800. The update for the GRIN is forward pass once. We'd be glad to address more.
>
> "The relation to prior MoE gating approaches (e.g., Expert Choice, Switch Transformers, or ST-MoE)..."
> - Thank you and the inhibitive gating can easily combine (it's a sigmoid activation and output multiplied with any layer output) with e.g. expert choice, switch trans, etc. The 'switch/choice/routing' gating is different from 'inhibitive' gating, and the latter easily integrates with the former. We add more explanation in the paper to clarify and formalize this important point.
>
> "The exposition is dense and occasionally ambiguous. Some notation and definitions ..."
> - Thank you for this important insight. We have made note of the ambiguity and density of our writing and have revised it in this new version. In particular, we re-wrote parts of related work and the experiments because they should be presented from an MTRL and RL perspective. We like the intent for the paper to be less ambiguous and have targeted the paper to be easily understood in this revision. We'd be glad to improve, so please let us know.
>
> Questions:
> "Can you provide variance reduction or gradient-norm analyses ..."
> - We'd be glad to provide more of this analysis with a full LSTM version where the theoretical analysis (as suggested above) could be well set up, so we the variance reduction or optimization improvements can be concretely designed. This is an insightful comment, while we may have to address it systematically.
>
> "What is the computational overhead (in FLOPs or wall-time) ..."
> - It is usually proportional to the size of the network 'after gating' if T=1 (e.g. +30% ) because of the interesting system insight that the network activations prior to gating can be cached.
>
> "Did you test GRIN in single-task RL or standard supervised settings ... "
> - Yes, this is very important. We performed experiments across multiple supervised tasks and language modeling. We found that GRIN (modulation, gating) works better when it's combined with effective use of MoE or parameter composition. So in a simple single supervised task, it wasn't very effective. While in a temporal dynamical system or sequence model, it works better (per LM results), and finally, MTRL, it seems to work very well. This is a highly interesting topic, while the comparison across modalities and across ML models may be the subject of more papers.
>
> "How sensitive is GRIN to the number of recurrence steps T..."
> - With the current definition of T, the algorithm isn't very sensitive to the recurrent step T, while we are interested in future work of versions of temporal definition or LSTM/GRU memory, where T could be more sensitive.

---

> > ### Author Response · Authors · 2025-12-01
> >
> > Continued:
> >
> > "Could GRIN be combined with routing regularizers (like load-balancing or entropy penalties) used in large-scale MoEs?..."
> > - Yes, GRIN as an architecture change does not add more regularization terms, so it fits well and can be applied directly with expert-balancing, expert selection, etc. For LM results, we didn't have the planned compute at to experiment with LLMs, while that's an important future work we'd like to explore. While it seems there are similar, simpler gating results with LLM's: https://openreview.net/pdf?id=1b7whO4SfY
> >
> > "What is the interpretation of negative correlations between ..."
> > - This was a scientific finding that surprised us as well. We are comfortable reporting that the primary gain from GRIN is from the modulation and gating of activations in MTRL, as opposed to changing routing weights in the MoE. This is consistent with findings here: https://openreview.net/pdf?id=1b7whO4SfY. It disproved a hypothesis that the gating control played a significant role in routing data in MoE for MTRL, but this may be different in other scenarios, such as LLMs with a large number of experts.
> >
> > "How is backpropagation-through-time truncated ..."
> > - Interestingly, we just use gradient norm clipping, and it works quite well. The T is also small, so we didn't encounter the need for truncation.
> >
> > "How does the recurrent inhibition affect gradient flow and credit assignment ..."
> > - Given the main contribution from modulation and gating (mainly correlations are less strong with mixture weights), we had less focus on the load-balancing and gradient flow. We observed that the experts are well-balanced, but think the generalization in MTRL may share parameters, and this will be an interesting topic when the data scale expands.
> >
> > "Is the recurrent head shared across layers or instantiated per MoE layer? Clarify parameter sharing and scaling implications."
> > - This is a very insightful comment. Currently, the recurrent hidden layer is shared across all the layers in the original network, because the empirical results from smaller-scale vision and text show that Global inhibition or diversity of connections helps. This is one of the key intuitions from cortical inhibitory neurons. We may experiment with a version that keeps the recurrent states separate per layer/section of the model, and possibly has a hierarchical structure. Thank you.
> >
> > I was wondering how GRIN behaves as the number of experts N grows (e.g., 8→64)? Is the global inhibition signal still efficient?
> > - We report results in ablation studies for growing expert numbers. The results on MT5 show that MTRL may share more parameters (on a small number of experts), so the GRIN signal's effectiveness for larger N is somewhat limited for Metaworld. This can also be interpreted from the correlation question above: since the GRIN signal is most effective for dynamic gating/modulation, the number of experts may not influence its efficiency too significantly.
> >
> > Have you tested inference latency when the recurrent inhibition loop is unrolled?
> > - Thank you for the question. We tested with batched inference and didn't perform stress-testing. At inference time, the caching also happens before the gating in the feedforward network, so the latency is linear with time-steps T while discounted by the proportion of the network that can be cached for its activations.

---

### Official Review · Reviewer_jc1L · 2025-11-01

**Soundness:** 2
**Presentation:** 1
**Contribution:** 2
**Rating:** 2
**Confidence:** 4

**Summary:**

Dynamic neural architectures, such as Mixture-of-Experts (MoE) models, are designed for efficiency and expressivity by using specialized pathways, but their core routing mechanism is challenging to design. To enhance this, the authors introduce Global Recurrent Inhibition Networks (GRIN), which are inspired by biological inhibitory neurons and apply an inhibitory gating function to the mixture weights, dynamic routers, and output representations. GRIN employs a recurrent algorithm that integrates signals from all neurons to maximize connection diversity, allowing the network to dynamically modulate its paths, suppress redundant signals, and improve pathway selectivity. The research tests a cascaded architecture where connection diversity progressively increases, demonstrating a direct correlation between this diversity and performance improvements. Empirical evaluations across simulated vision, language modeling, and MTRL show that GRIN consistently outperforms baseline MoE architectures, particularly on the challenging MetaWorld MT10 and MT50 benchmarks.

**Strengths:**

- I believe the introduced solution is interesting and the idea of considering a global state for routing sounds.
- The methodology section is well-presented as well as the literature discussion.

**Weaknesses:**

- The paper needs a couple more iterations over the writing and formatting. For instance, there are many unused spaces (which created noticeable white spaces at every page). In addition, subsection heading 5.2 is placed in a wrong position.
- However, the main and the most crucial limitation of this work is the motivation of the presented problem and solution to MTRL. I think this point is not well stated in the work. I was also confused because of the cross domain experiments. It seems like the solution is for many domains (which is totally fine), why should the paper target MTRL as the main problem?
- The authors choose to not run the baselines (including MOORE) on MT10 while rerunning only MOORE on MT50. Why is that? I think it is better to either borrow the results or rerun everything.
- In addition, there is inconsistency in training steps across baselines in MT50, the authors acknowledged that. But this is an indication that this submission is incomplete.
- I believe the performance gain of using the method in MTRL setting is not significant.
- I think the discussion in Section 4.2 needs writing adjustment since I could not understand everything.

I think this work has a great potential. However, the main issue with this submission is that the motivation for the problem and solution in the context of MTRL, which is the main problem setting, is not well highlighted and written. In general, the submission needs some writing improvements. In addition, the main results in the paper, on Metaworld, should be completed. Considering all of that in the submission, the work would be ready. However, I understand that the rebuttal period could be not enough for that.

**Questions:**

- What is the motivation behind proposing this solution for MTRL? Why is it important to consider a global state to create pathways in MTRL?
- What is the reason for rerunning only MOORE in MT50  while taking results from the MOORE paper for others? I would recommend rerunning all baselines or at least becoming more consistent in terms of results reporting. This should improve the quality of the paper.
- Since it is a MTRL paper, why not focusing on more MTRL benchmarks, for example, by reporting more results on the MiniGrid benchmark instead of the cross domain studies? Maybe moving them to the appendix or highlighting only one. Yet, I believe there is no real space issue since there are white spaces at the end of each page.
- How important is to use MOORE, in terms of orthogonal features, as a base algorithm?

---

> ### Author Response · Authors · 2025-12-01
>
> Thank you, reviewer  jc1L for your constructive and highly insightful comments!
>
> "The paper needs a couple more iterations ..."
> - Please kindly see the revision as we tried to improve the writing with a few more iterations. The misplacement of figures/tables have been fixed. Before, the margins were oversized due to a hidden issue from the 'geometry' package.
>
> "However, the main and the most crucial limitation of this work is the motivation of the presented problem and solution to MTRL..."
> - To address your comment, we realized the presentation impact was limited because it overlooked the RL and MTRL research perspective. Rather than presenting to a generic deep learning audience, we like the RL angle of diversification and parameter composition, and we rewrote the abstract and introduction to explain why dynamic gating, modulation across time, is so important for complex dynamical systems in MTRL. This is as important as transformers/LLMs, if not more. Please let us know if this revision addresses your concern, and we take this feedback very seriously. Thank you.
>
> "The authors choose to not run the baselines (including MOORE) on MT10 ..."
> - Thanks for your comment here, while I'd like to point out we did in fact run the baselines on MT10. If you'd kindly compare Table 1 with the MOORE paper, you'd see that the per-epoch numbers are different.
>
> "In addition, there is inconsistency in training steps across baselines in MT50, the authors acknowledged that...."
> - Thank you! We eventually did, in fact, finish the 100M env. steps with MOORE and GRIN because we present now a stat. sig. improvements of +3.7% with GRIN! While the expert opinion from https://openreview.net/pdf?id=1de3azE606 might orient us to use V2 rewards and we thank all reviewers and experts for your kind help.
>
> "I believe the performance gain of using the method in MTRL setting is not significant."
> - Thank you - with IQM definition suggested by reviewer eCJY, we could see the MT50 stat-sig results, and MT10 (to a lesser extent, but still stat. sig.)
>
> "I think the discussion in Section 4.2 needs writing adjustment since I could not understand everything."
> - The writing is revised now, and feel free to raise further feedback - we'd be glad to address.
>
> "I think this work has great potential. However, the main issue with this submission is that the motivation for the problem...I understand that the rebuttal period may not be enough for that."
> - Thank you for your kind words. Upon discovering the effectiveness of orthogonalization in MOORE, we've tried very hard to address your comment on the motivation because we also think it's one of the key connections we could make with RL researchers. It is more important for the GRIN algorithm to be able to address key concerns and scientific questions in RL, and I hope our revision shows that effort. We'd be glad to do more work on this. Thank you. For the full results, we did manage to finish the 100M steps, also test out the repository: https://github.com/rainx0r/metaworld-algorithms suggested by reviewer eCJY. We now focus on the improvements with GRIN because time and resources are critical for ICLR. Given the incredible results from MOORE and its effectiveness, we hope that you could also kindly understand our intense effort and the key results presented per your request.
>
> Questions:
>
> "What is the motivation behind proposing this solution for MTRL? Why is it important to consider a global state to create pathways in MTRL?"
> - It is utmost important to apply dynamic gating with temporal context to the most complex dynamical systems in MTRL. Due to MTRL's need for parameter composition (MoE) to generalize across tasks, dynamic gating can be most effect or consequential when applied to MTRL. As illustrated in https://openreview.net/pdf?id=1b7whO4SfY dynamic gating in MoE's is a critical topic that can significantly impact performance and optimization in sequential and RL models.
>
> "What is the reason for rerunning only MOORE in MT50 ...? I would recommend rerunning all baselines ... improve the quality of the paper."
> - Thank you for your comment. We did rerun results for MT10, MT5 in Metaworld, and also for MiniGrid because of the reporting consistency it brings. As you may know, before knowing the JAX implementation, our resources were constrained. We would definitely like to do more of that to improve the quality of the paper.
>
> "Since it is a MTRL paper, ... by reporting more results..."
> - Thank you, and yes we reported some MiniGrid results in the paper, and we would like to report more benchmarks, while the consistency in benchmarks (e.g. env. steps) made our decision a bit more conservative.
>
> "How important is to use MOORE, in terms of orthogonal features, as a base algorithm?"
> - We found it highly effective, especially the G-S process which diversifies the representations. Performance is also quite consistent in our experiments.

---

### Author Response · Authors · 2025-12-01
**Rebuttal revision and thank you.**

We thank the reviewers for your helpful and constructive feedback.

We have respectfully provided the rebuttal revision of the paper, with changes highlighted in blue. Kindly read it, and we'd appreciate your comments in the short time remaining in the rebuttal period. It is our pleasure to call to your attention one of the NeurIPS 2025 best papers: https://openreview.net/pdf?id=1b7whO4SfY as an important additional reference to our approach.

As first-time researchers on Metaworld, we thank reviewers and researchers for helping us align results on Metaworld, Metaworld+, and get us up-to-speed with SOTA implementations. We plan to continue to contribute and help improve, align the MTRL benchmark for openness and reproducibility, while working with reviewers to improve the paper. Before this, we hope to bring to your attention that it is challenging and it could be confusing to align protocols and reproduce results on Metaworld MT50. Our experience below shows how it could have been very expensive for researchers to do work on the Metaworld MT50 dataset.

- Prior to Metaworld+ and https://github.com/rainx0r/metaworld-algorithms, we rented 30 A100 GPUs, and finishing the runs  on MT50 took about a month's machine time, with compute/storage costs over 30k
- With the MOORE JAX implementation in https://github.com/rainx0r/metaworld-algorithms with reward function V2, we experimented during the rebuttal period. After fixing minor details such as alpha=1e-8 in Adam, we arrived at around 67% success rate on average at 100M (1e8) environment steps
For example: \textcolor{blue}{total_steps=99999950, mean evaluation success rate: 0.6708 return: 223.6119}
- We acknowledge it could be challenging to align in the number of environment steps to reproduce MOORE, across (1) authors' emails (asking us to run over 100 epochs in MOORE code, which is over 500M environment steps); (2) the moore GitHub code results in 56% in 100M env. steps or 20 epochs, (3) the Metaworld+ paper also first reported 65-68% MOORE result in arxiv versions, although it eventually reported 72% results for MOORE, we have been looking in the paper for how many env. steps it took and how to arrive at the number.

In our current revision, we can report all the factual results we obtained using the MOORE code base: https://github.com/AhmedMagdyHendawy/MOORE . These are the results we could commit to and report as  researchers within the time frame given. We did, in fact, finish the 100M experiments on MT50 with 10 random seeds as promised, and GRIN showed +3.7% IQM and statistically significant improvements against the MOORE baseline. As we had followed the author's email, we had the experiment run to 250M env. steps (50 epochs) before the cost was too prohibitive, it didn't reach 72% reported numbers, but the GRIN improvements sustained and remained stat-sig. For future work, we will leverage https://github.com/rainx0r/metaworld-algorithms

We ask the reviewers, meta-reviewer, or AC to kindly review our results as reported factually. Our intention is to prove out the GRIN algorithm, with the belief that the MoE and complex temporal dynamical systems in MTRL are the best demonstration of the dynamic gating algorithm, with reference to the rebuttal comments we post. Thank you very much.

---

### Note · Authors · 2025-12-04

I have read and agree with the venue's withdrawal policy on behalf of myself and my co-authors.